# Genetic and Molecular Characterization of H9c2 Rat Myoblast Cell Line

**DOI:** 10.3390/cells14070502

**Published:** 2025-03-28

**Authors:** Thomas Liehr, Stefanie Kankel, Katharina S. Hardt, Eva M. Buhl, Heidi Noels, Diandra T. Keller, Sarah K. Schröder-Lange, Ralf Weiskirchen

**Affiliations:** 1Institute of Human Genetics, Jena University Hospital, Friedrich Schiller University, D-07747 Jena, Germany; stefanie.kankel@med.uni-jena.de; 2Institute of Molecular Pathobiochemistry, Experimental Gene Therapy and Clinical Chemistry (IFMPEGKC), RWTH University Hospital Aachen, D-52074 Aachen, Germany; khardt@ukaachen.de (K.S.H.); dikeller@ukaachen.de (D.T.K.); saschroeder@ukaachen.de (S.K.S.-L.); 3Electron Microscopy Facility, Institute of Pathology, RWTH University Hospital Aachen, D-52074 Aachen, Germany; ebuhl@ukaachen.de; 4Institute for Molecular Cardiovascular Research (IMCAR), RWTH University Hospital Aachen, D-52074 Aachen, Germany; hnoels@ukaachen.de; 5Department of Biochemistry, Cardiovascular Research Institute Maastricht, Maastricht University, 6211 Maastricht, The Netherlands

**Keywords:** cardiomyoblast, myocardium, in vitro model, SKY analysis, STR profiling, karyogram, next-generation sequencing, cell authentication, ICLAC

## Abstract

This study presents a comprehensive genetic characterization of the H9c2 cell line, a widely used model for cardiac myoblast research. We established a short tandem repeat (STR) profile for H9c2 that is useful to confirm the identity and stability of the cell line. Additionally, we prepared H9c2 metaphase chromosomes and performed karyotyping and molecular cytogenetics to further investigate chromosomal characteristics. The genetic analysis showed that H9c2 cells exhibit chromosomal instability, which may impact experimental reproducibility and data interpretation. Next-generation sequencing (NGS) was performed to analyze the transcriptome, revealing gene expression patterns relevant to cardiac biology. Western blot analysis further validated the expression levels of selected cardiac genes identified through NGS. Additionally, Phalloidin staining was used to visualize cytoskeletal organization, highlighting the morphological features of these cardiac myoblasts. Our findings collectively support that H9c2 cells are a reliable model for studying cardiac myoblast biology, despite some genetic alterations identified resembling sarcoma cells. The list of genes identified through NGS analysis, coupled with our comprehensive genetic analysis, will serve as a valuable resource for future studies utilizing this cell line in cardiovascular medicine.

## 1. Introduction

H9c2 is a cell line derived from the embryonic rat heart, established in 1976 by B. W. Kimes and B.L. Brand [1]. It is widely used in cardiovascular research due to its ability to differentiate into skeletal and cardiac myocytes, with the latter induced by supplementation of all-trans-retinoic acid [2]. This cell line retains many characteristics of cardiac muscle cells, including the expression of cardiac-specific proteins such as troponin T and myosin heavy chains [3,4], making it an excellent model for studying heart development, function, and disease mechanisms, particularly in the context of hypertrophy and ischemia-reperfusion injury. Additionally, H9c2 cells are often employed in drug testing and toxicology studies due to their responsiveness to various pharmacological agents and their capacity for cellular signaling studies related to heart health and disease [5,6,7,8].

In addition to troponin T and myosin heavy chains, H9c2 cells express transcription factors such as GATA4 and Nkx2.5, which are crucial for cardiac development and function [9,10]. H9c2 cells have been studied for their response to stressors like hypoxia or oxidative stress [11,12], revealing insights into the molecular pathways involved in cardiomyocyte survival and apoptosis. Furthermore, researchers have utilized techniques such as CRISPR/Cas9 gene editing to manipulate specific genes within H9c2 cells to study their roles in cardiovascular diseases [13,14].

Overall, H9c2 has become a valuable tool to gain insights into cardiac biology and pathology. However, it is essential to consider its limitations as a model system when translating findings to in vivo situations. While H9c2 cells share many characteristics with primary cardiomyocytes, they may not fully replicate the complex genetic and functional properties of adult heart tissue and are considerably different from both primary neonatal cardiomyocytes and adult myocardium [15]. Therefore, findings from studies using this cell line should be interpreted with caution when considering their relevance to in vivo conditions.

Moreover, while H9c2 cells are widely used in cardiovascular research, it is important to note that their specific genetic characteristics and a standardized short tandem repeat (STR) profile have not been thoroughly established. This lack of comprehensive genetic characterization raises concerns regarding the potential for genetic drift or variability within the cell line, which could impact experimental reproducibility and reliability. Researchers using H9c2 cells should be aware of this limitation and consider verifying the identity of their cell lines through additional methods, such as DNA fingerprinting or other genomic analyses, to ensure consistency and accuracy in their studies.

In this study, we conducted a comprehensive genetic analysis of H9c2 cells. We established a karyotype and utilized multicolor fluorescence in situ hybridization (mFISH) to clarify the chromosomal makeup. Additionally, we established an STR profile with 31 species-specific markers to verify the identity and stability of the cell line. We also examined the transcriptome through mRNA sequencing (mRNA-Seq) using next-generation sequencing (NGS) and valuated the characteristic morphological traits of this rat heart cell line through electron microscopy, Western blotting, and Phalloidin staining.

## 2. Materials and Methods

### 2.1. Literature Search

The PubMed database was searched for papers that used H9c2 cells. The search was performed using the specific term “H9c2” to retrieve articles. No filters were applied, allowing for the identification of all studies related to the use of H9c2 cells in various experimental contexts.

### 2.2. Cell Culture

The rat cell line H9c2 was obtained from ATCC (CRL-1446) and cultured in Dulbecco's Modified Eagle’s Medium (DMEM, high glucose #D6171, Sigma-Aldrich, Merck, Taufkirchen, Germany) containing 1.5 g/L sodium bicarbonate, supplemented with 10% fetal bovine serum (#F7524, Sigma-Aldrich), 4 mM L-glutamine (#G7513, Sigma-Aldrich), and 1× penicillin/streptomycin (DE17-602E, Lonza, Cologne, Germany). The medium was changed every 2–3 days, and cells were split using Accutase^®^ solution (A6964-100ML, Sigma-Aldrich) at a ratio of 1:3 when they reached confluence. All experiments were carried out at passages 2–4 after receiving the cells from ATCC. The cells were maintained at 37 °C in an environment with 95% air and 5% CO_2_.

### 2.3. Mycoplasma Testing

The identification of possible Mycoplasma species contaminations in cell culture supernatants was performed utilizing the Venor^®^GeM OneStep kit (#11-8050, Minerva Biolabs GmbH, Berlin, Germany) following the manufacturer’s instructions. Briefly, 2 µL of fresh medium (medium (-)), 2 µL of supernatant from H9c2 cell cultures, or 2 µL of the positive control provided with the kit were used for PCR. The PCR protocol included an initial cycle at 94 °C for 2 min, followed by 39 cycles at 94 °C for 30 s, 55 °C for 30 s, and 72 °C for 30 s, ending with a cooling phase at 4 °C. Subsequently, the PCR products were analyzed on a standard 2% agarose gel in 1× TAE buffer consisting out of 40 mM Tris, 20 mM acetic acid, and 1 mM EDTA (pH 8.6) for 50 min at 90 V, with ethidium bromide added for visualization. The resulting amplicons were then examined using a standard gel imaging system (Intas Science Imaging GmbH, Göttingen, Germany).

### 2.4. Short Tandem Repeat (STR) Profiling

The STR profiling and evaluation for interspecies contamination in H9c2 cells were performed using the cell line authentication service offered by IDEXX (Kornwestheim, Germany) through the CellCheck^TM^ Rat system. This system employs a dinucleotide repeat assay to generate a genetic profile of the cells, consisting of 31 unique STR markers that are specific to different species.

### 2.5. Preparation of H9c2 Metaphase Chromosomes, Karyotyping, and Molecular Cytogenetics

Chromosomes from H9c2 cells were prepared following a standard protocol for metaphase preparation, with some modifications [16]. H9c2 cells were incubated at 37 °C in T25 flasks until they achieved a semi-confluent state. After treatment with KaryoMAX colcemid solution (#15212012, Gibco, ThermoFisher Scientific, Schwerte, Germany), the cells were detached from the flask surface using a gentle trypsin–EDTA solution (#T4174, Sigma-Aldrich) and collected in a centrifuge tube. Following a brief centrifugation step, the cells underwent hypotonic treatment with 0.56% KCl for 30 min at 37 °C before being fixed in a mixture of acetic acid and methanol (1:3). Chromosome spreads were then air-dried from the fixed cell suspension and used for mFISH as described before [17]. To detect interchromosomal rearrangements with the chromosomes of H9c2, the commercially available rat probe set 22xRat (MetaSystems, Altlussheim, Germany) was employed. It is important to note that due to the specific design of this probe set, chromosomes 13 and 14 cannot be distinguished after FISH analysis [18]. For analysis, 30 metaphases were examined using a Zeiss Axioplan microscope equipped with the ISIS software package, version 6.1.1, MetaSystems). To create standard chromosome banding patterns, metaphases were sorted according to FISH signals and counterstained with 4′,6-diamidino-2-phenylindole (DAPI), then transformed into an “inverted DAPI-banding” pattern with a single click in the software used.

### 2.6. Virtual Comparative Genomic Hybridization

By analyzing the data obtained from mFISH along with inverted DAPI-banding, we were able to identify approximate losses and gains of chromosomal regions in rat chromosomes. These regions were subsequently mapped to their estimated molecular positions based on Rat RGSC 5.0/rn5 using the UCSC genome browser. Employing the “View, In Other Genomes” feature allowed us to translate these regions into the human genome (build: GRCh37/hg19) and determine homologous regions with corresponding gains or losses.

### 2.7. Next-Generation Sequencing and Data Analysis

High-molecular-weight cellular RNA from five 100 mm^2^ plates of H9c2 cells, which were grown to 80% confluence, was isolated using a CsCl_2_ density gradient centrifugation protocol [19]. The concentration, purity, and quality of the purified RNA were assessed using standard UV spectroscopy and the Agilent 4200 TapeStation platform (Agilent Technologies Inc., Waldbronn, Germany). After depleting ribosomal RNA, mRNA was converted into a sequencing library using the NEBNext Multiplex Oligos for Illumina Index Primers Set 1 kit. Sequencing was performed on the Illumina platform (Illumina Inc., San Diego, CA, USA) with pre-filled cartridges (MiSeq Reagent kit V2, 300-cycles) from Illumina, and the results were converted into FASTQ data files. The cDNA library construction and sequencing took place at the IZKF Genomic Facility of the University Hospital Aachen. FASTQ files were generated using bcl2fastq (Illumina) before downstream analyses. Samples were processed using the nf-core/RNA-seq pipeline version 3.12 [20] in Nextflow 23.10.0 [21]. Lane-level reads were trimmed with Trim Galore 0.6.7 [22], aligned to the *Rattus norvegicus* (Rnor_6.0) reference transcriptome using STAR 2.7.9a [23], and quantified at gene and transcript levels with Salmon v1.10.1 [24], resulting in length-normalized Transcripts Per Million (TPM) values.

### 2.8. Electron Microscopic Cell Analysis

Electron microscopic analysis of H9c2 cells was performed following established protocols [25]. Cells were fixed in 1× phosphate-buffered saline (PBS) with 3% glutaraldehyde, washed with 0.1 M Soerensen’s phosphate buffer, and post-fixed in 1% osmium tetroxide. Dehydration was achieved using a series of ethanol solutions (30% to 100%), followed by incubation in propylene oxide and Epon resin mixtures, which were polymerized at 90 °C for two hours. Ultrathin sections (90–100 nm) were cut, stained with uranyl acetate and lead citrate, and examined with a Zeiss Leo 906 transmission electron microscope (Carl Zeiss AG, Oberkochen, Germany) at 60 kV. Images were captured at specified magnifications (2156×, 10,000×, 21,560×, and 27,800×), respectively.

### 2.9. Western Blot Analysis

Protein extracts, quantification, and Western blot analysis were performed following established protocols. Protein samples (50 µg/lane) were heated at 80 °C for 10 min and separated using 4–12% Bis-Tris gels (Invitrogen, Thermo Fisher Scientific, Schwerte, Germany) under reducing conditions with 2-(*N*-morpholino)ethanesulfonic acid (MES) running buffer. The proteins were transferred onto a 0.45 µm nitrocellulose membrane (#GE10600002, Amersham^TM^ Protran^®^ Western-Blotting Membranes, Merck), with transfer efficiency verified by Ponceau S staining. Blocking was carried out in Tris-buffered saline containing 0.1% Tween 20 and 5% non-fat milk powder. The membranes were incubated with primary antibodies and detection was achieved using horseradish peroxidase-conjugated secondary antibodies and chemiluminescence (Supersignal™ West Dura extended duration substrate, #34076, Thermo Fisher Scientific, Schwerte, Germany). As a positive control, we used protein extracts generated from the heart of a female rat that was homogenized in an MM400 mixer mill (Retsch GmbH, Haan, Germany) using an established protocol [26]. Details of the antibodies used in our study are provided in Table 1.

### 2.10. Phalloidin Stain

Microfilament staining was conducted according to previously established methods [27]. In summary, 30,000 H9c2 cells were plated on glass coverslips placed in a 24-well plate. After a 48 h incubation period, the medium was removed, and the cells were rinsed with phosphate-buffered saline (PBS) before being fixed in 3.7% paraformaldehyde (pH 7.4) for 20 min in the dark. Subsequently, the cells were permeabilized using a precooled solution of 0.1% sodium citrate and 0.1% Triton X-100 for 3 min on ice. After additional washes in PBS, nonspecific binding sites were blocked with PBS containing 50% fetal bovine serum (FBS) and 0.5% bovine serum albumin (BSA) for one hour at room temperature. Under exclusion of light, the cells were then stained with an Alexa Fluor™ 488 Phalloidin conjugate for 20 min, followed by nuclear counterstaining with 200 ng/mL DAPI solution (#D1306, Thermo Fisher Scientific) for 15 min. Finally, samples were mounted using PermaFluor™ aqueous mounting medium (#TA-030-FM, Thermo Fisher Scientific) and observed under a Nikon Eclipse E80i fluorescence microscope equipped with the NIS-Elements Vis software package (version 3.22.01). For detailed protocol instructions, please refer to [27].

## 3. Results

### 3.1. Usage of H9c2 Cells in Biomedical Research

H9c2 cells play a crucial role in biomedical research, especially in the study of cardiovascular diseases and cardiac physiology. Their ability to differentiate into cardiomyocyte-like cells makes them a valuable model for investigating various aspects of heart function, drug responses, and disease mechanisms [2]. The widespread use of H9c2 cells is evident by their inclusion in 7334 studies listed on PubMed as of 12 February 2025. This extensive research highlights the importance of H9c2 cells in advancing our understanding of cardiac biology and developing therapeutic strategies for heart-related conditions.

### 3.2. Phenotypic Appearance of H9c2 Cells

The phenotypic characteristics of H9c2 cells, as observed through light microscopy, show a distinct morphology typical of cardiac progenitor cells (Figure 1). These cells often have a fibroblast-like appearance with elongated and spindle-shaped structures, allowing them to form a dense and flat monolayer when cultured. Under optimal growth conditions, H9c2 cells exhibit prominent cytoplasmic extensions and can sometimes cluster together, indicating their ability to spontaneously differentiate into cardiomyocyte-like cells.

### 3.3. Electron Microscopic Analysis of H9c2 Cells

When observing H9c2 cells through electron microscopy, it becomes apparent that these cells have a well-developed cytoplasm abundant in mitochondria, vacuoles, lipid droplets, and lysosomes, indicating high metabolic activity (Figure 2). The presence of a significant amount of rough endoplasmic reticulum (rER) suggests a high level of protein synthesis, supporting the metabolic demands of these cells.

### 3.4. Expression of Typical Fibroblast Markers in H9c2 Cells

#### 3.4.1. Next-Generation Sequencing

To analyze the expression profile of H9c2 cells without bias, we conducted next-generation sequencing (NGS) mRNA sequencing. This method provided a comprehensive overview of the gene expression in this cell line. Our investigations revealed a diverse gene expression profile that indicates their cardiac lineage and functional potential. Notably, we found the expression of various members of the myosin family, which play critical roles in muscle contraction and cellular motility (Table 2).

The presence of these myosin genes suggests that H9c2 cells retain key features associated with cardiac muscle function, highlighting their potential as an excellent model for studying contractile mechanisms and heart development. Furthermore, the differential expression of specific myosin isoforms may provide insights into the regulatory pathways governing cardiomyocyte differentiation and maturation.

In our NGS analysis of H9c2 cells, we identified several additional classical markers of cardiomyocytes, further confirming their cardiac lineage and functional properties. Among the marker genes detected were genes that are known to be either cardiomyocyte-specific or functionally relevant in cardiomyocytes. Examples include *Actn1*, *Actc1*, *Adipoq*, *Adipor1*, *Adipor2*, *Alcam*, *Ankrd1*, *Atp2a2*, *Anxa5*, *Anxa6*, *Bdnf*, *Bmp4*, *Cav2*, *Cav3*, *Cdh2*, *Cpt1a*, *Csrp2*, *Ctnnb1*, *Des*, *Dmd*, *Eno3*, *Fabp3*, *Fgf2*, *Fhl2*, *Gata-4*, *Gata-6*, *Gja1*, *Hand2*, *Il11ra*, *Igf1*, *Lox*, *Mef2c*, *Mfn2*, *Mitf*, *Mov10l1*, *Notch1*, *Nkx2-5*, *Pde1a*, *Pde4d*, *Pygm*, *Rgd1565355* (CD36), *Ryr2*, *Sgpl1*, *Tnnc1*, *Tnni3*, *Tnnt1*, *Tnnt2*, *Tpm1*, *Trpv1*, *Ttn*, and *Pnmt* (Table 3). The expression of these genes indicates key processes involved in cardiac development and function. This comprehensive gene expression profile highlights the utility of H9c2 cells as a relevant model for investigating cardiomyocyte biology and offers insights into the molecular mechanisms underlying heart physiology and pathology.

However, we found no expression of the natriuretic peptides *Nppa* (atrial natriuretic factor, ANF), *Nppb* (brain natriuretic peptide, BNP), and *Nppc* (natriuretic peptide, type C, CNP) genes, which are abundantly expressed in the atrial and ventricular myocardium during embryonic and fetal stages and are relevant in cardiac remodeling (Appendix A) [29,30].

Additionally, there were no transcripts found for *Pou5f1* (OCT4), a factor of pluripotency that drives the dedifferentiation of adult cardiomyocytes into a fetal state [31]. The transcription factor *Hand1* (eHAND), marking cardiac progenitor cells, was also not expressed in H9c2 (Appendix A) [32]. In contrast, the closely related *Hand2* (dHand), which alone is sufficient to promote differentiation onset, was expressed in H9c2 cells [33]. Furthermore, *Fgf23* and *Cxcr4*, typically expressed by cardiac myocytes and important for cardiogenesis [34,35] were not found to be expressed in H9c2 cells.

#### 3.4.2. Analysis of Protein Expression and Cytoskeletal Organization in H9c2 Cells

Our NGS data of H9c2 cells has shown the expression of a variety of genes previously associated with cardiomyocyte functionality. In addition to those mentioned above, we identified the expression of vinculin (*Vcl*) [36], β-catenin (*Ctnnb1*) [37], vimentin (*Vim*) [38], β-actin (*Actb*) [39], four and a half LIM domain 2 (*Fhl2*) [40], fibronectin (*Fn1*) [41], collagen type I α 2 chain (*Col1a2*) [42], AKT serine/threonine kinase 1 (*Akt1*) [43], tubulin α1A (*Tuba1*) [44], α-smooth muscle actin (*Acta2*) [45], BCL2-associated X (*Bax*) [46], cytochrome c (*Cycs*) [47], collagen type III α1 (*Col3a1*) [42], heat shock protein 90 (*Hsp90aa1*) [48], beclin 1 (*Becn1*) [49], gap junction protein α1 (*Gja1*) [50], and ferritin heavy chain 1 (*Fth1*) [51] (Appendix A).

Western blot analysis confirmed the expression of all these genes in H9c2 cells (Figure 3). Specifically, vinculin, vimentin, β-actin, fibronectin, collagen type I, AKT, Bax, collagen type III, and beclin 1 showed increased expression in H9c2 cells compared to their expression in total heart cell extracts from female rats. Additionally, we found protein expression of actinin, alpha 1 (Actn1), but no expression of actinin, alpha 2 (Actn2). These findings align with the NGS data (Table 3). Furthermore, troponin I, which was expressed at overall low quantities in NGS (0.52 TPM), and troponin T, which was expressed in similar mRNA quantities to Actn1, was undetectable at the protein level in H9c2 cells. However, these proteins were present in extracts of heart tissue, confirming the functionality of the antibodies used.

One characteristic feature of cardiomyocytes is the presence of a non-contractile, densely packed cytoskeleton composed of cytoplasmic actin, microtubules, and intermediate filaments. This cytoskeleton plays crucial roles in the electrical and mechanical coupling of cardiomyocytes [39].

Specifically, actin is a vital component of sarcomeres in cardiomyocytes that can transition between a monomeric (G-actin) and a polymeric filamentous (F-actin) form. Staining H9c2 cells with Alexa Fluor 488^TM^-labeled Phalloidin conjugate resulted in the labeling of F-actin filaments, displaying the typical bundle morphology of these large and flat growing cells (Figure 4).

### 3.5. Karyotype Based on Molecular Cytogenetic Analyses

The karyotype of the rat cell line H9c2 reveals a chromosomal composition characterized by significant aneuploidy and structural abnormalities (Figure 5).

The karyogram analysis revealed a chromosome count ranging from 74 to 81, indicating a triploid status (3n), commonly seen in transformed cell lines. Among the chromosomes, two X chromosomes are present, with one showing notable deletions between bands q2 and q3. Additionally, there is an extra copy of chromosome 1, and a small derivative of chromosome 1 with loss of the entire long arm distal from subband q12. In addition to three copies of chromosome 2, there is an additional derivative of chromosome 2 with the loss of the entire long arm starting at subband q13. Notably, there are no normal chromosomes; instead, three additional derivatives of chromosome 12 are present, consisting predominantly of the short arm of chromosome 12 and the long arm of chromosome 3. The karyotype also includes three copies of chromosome 20 material. However, one chromosome 20 underwent a fission event, leading to two derivative chromosomes 20: one consisting of the long arm and one of the short arm of a normal chromosome 20. One extra copy each of chromosomes 6, 9, and 14 is present. Finally, a derivative chromosome 12 involving both the X chromosome and chromosome 12 was identified.

Overall, this karyotype reflects the genetic changes that occur during the adaptation or transformation processes typical for H9c2 cells. These chromosomal abnormalities may have implications for their behavior in research contexts, particularly in studies related to cardiac function or disease models.

### 3.6. Virtual Comparative Genomic Hybridization

We conducted virtual Comparative Genomic Hybridization (vCGH), a powerful tool for analyzing genomic alterations in cell lines. The motivation behind performing vCGH in H9c2 cells stems from the need to gain a deeper understanding of the genetic landscape and chromosomal abnormalities, such as gains, losses, and structural rearrangements, that contribute to the cellular characteristics of H9c2 cells. These alterations may impact their behavior and responsiveness in experimental settings.

The results of the vCGH analysis on H9c2 cells reveal significant genomic alterations, including both gains and losses across various chromosomes (Table 4).

Starting with the gains, there is a notable duplication in the region from 1pter to 1q12, which corresponds to several human chromosomal regions, including 6q22.31 to q27 and others, indicating a complex gain involving multiple loci. Additionally, there is an increase in copy number from 1q12 to 1qter, suggesting further amplification in this chromosome segment that spans multiple human regions, such as 10q23.2 to q26.3 and others. The analysis also identified gains on chromosome 2, with a specific increase noted from 2pter to q13. Chromosome 6 exhibits a gain from its short arm (p) to the long arm (q), while chromosome 9 shows an overall gain from pter to qter. Furthermore, there is a significant amplification involving chromosome 12, where four additional copies were detected in the region extending from pter to q12.

Losses are equally prominent in this karyotype; specifically, there is a substantial deletion observed on chromosome 3 from pter to q11, resulting in three missing copies. Additionally, the X chromosome displays a loss between bands Xq2 and Xq3.

Nevertheless, the visualization of the vCGH analysis demonstrates that the gains in this cell line are more prominent than the losses (Figure 6).

Most cardiac cancers in humans are secondary, with intimate and undifferentiated sarcomas being the most common primary ones. Interestingly, amplification of *MDM2* (12q15), *MDM4* (1q32.1), and *CDK6* (7q21.2) is frequently observed, along with *PDGFRA* (4q12), *CDK4* (12q14.1), and *TERT* (5p15.33) in humans [54]. In the H9c2 cell line, an increase in copy numbers is seen in regions containing *CDK6* (four additional copies), *PDGFRA* (one additional copy), and *TERT* (two additional copies). The loss of 9q material is also associated with human sarcomas [55,56].

These results indicate significant chromosomal rearrangements within H9c2 cells, which could be a hint at the evolution of the original embryonic rat heart cells toward malignant, potentially immortal cells similar to human sarcoma. The methylation pattern of H9c2 cells may also be similar to that of intimal and undifferentiated cardiac sarcomas, as described by [54]. Additionally, mutation analyses in *MDM2* (12q15), *MDM4* (1q32.1), *CDK4* (12q14.1), and *TERT* may be of interest to check for known mutations with oncogenic potential.

### 3.7. Short Tandem Repeat Analysis

Subsequently, we conducted STR profiling on H9c2 cells to assess their genetic stability and establish a marker panel for verifying the authenticity of this cell line. It is important to note that despite the widespread use of H9c2 cells in cardiovascular research, there is no published STR profile available for them, highlighting the significance of our analysis. We employed a comprehensive panel of 31 markers located across 20 autosomes of the rat genome for the STR profiling (Table 5).

The STR profile we have established for H9c2 cells is unique and distinctly differs from those of other rat cell lines, such as CFSC-2G, HSC-T6, PAV-1, and Rat-1, which we reported previously [19,25,57,58]. This differentiation highlights the genetic uniqueness of H9c2 cells and reinforces their identity as a distinct cell line.

## 4. Discussion

In this study, we conducted a comprehensive genetic characterization of the H9c2 cell line, commonly used in cardiovascular research. A significant contribution of our work is establishing a unique STR profile for H9c2 cells. This genetic fingerprint enables quick and reliable confirmation of the identity of this cell line, preventing issues related to misidentification or cross-contamination in biomedical research. It also helps assess the genetic stability of the cell line and addresses previous concerns about potential genetic drift, which can affect experimental reproducibility [59].

Additionally, our karyotyping and molecular cytogenetic analyses revealed complex chromosomal characteristics typical of transformed cell lines. We identified significant chromosomal alterations in H9c2 cells, including aneuploidy and structural abnormalities common in transformed cell lines. The karyotype showed chromosome counts ranging from 74 to 81, indicating a triploid status. Notable findings included deletions on chromosomes X and 2, and additional copies of chromosomes 1, 6, and 9. We also observed derivative chromosomes from translocations involving chromosomes 3 and 12. These chromosomal changes may impact the cellular behavior of H9c2 cells, affecting their proliferation, differentiation potential, and response to experimental conditions. Understanding these genetic changes is crucial for accurately interpreting research outcomes when using H9c2 cells in cardiovascular studies.

Chromosomal changes during routine cell culture can influence cellular behavior, including proliferation, differentiation, and response to stressors [60]. Genetic factors play a crucial role in cardiovascular research, where cellular responses and disease outcomes can vary significantly based on the genetic context and cell state [61,62].

NGS analysis and Western blot analysis provided valuable insights into the transcriptomic landscape of H9c2 cells. Our findings indicate that these cells retain key features associated with cardiac lineage, as shown by the expression of various myosin genes and other cardiomyocyte-specific markers. These findings align with previous proteomic investigations of H9c2 cells, suggesting that this cell line is a suitable model for very immature myogenic cells with skeletal muscle commitment [15]. The determined gene expression profile can now be used by researchers to analyze molecular pathways and identify potential key regulatory networks in H9c2 cells.

Nevertheless, despite their advantages as an in vitro model system, it is essential to acknowledge certain limitations associated with H9c2 cells. While they exhibit many characteristics similar to primary cardiomyocytes, they do not fully replicate the complexity of adult heart tissue or its microenvironment. For instance, our analysis showed a lack of expression for natriuretic peptides, markers typically abundant in mature cardiomyocytes, which may affect their utility in specific applications related to heart failure or hypertrophy studies [63]. Consequently, findings derived from H9c2 studies should be interpreted cautiously when extrapolating to in vivo conditions.

We should acknowledge that our NGS data were obtained from H9c2 cells cultured under basal conditions without the addition of substances such as retinoic acid, which are known to drive the differentiation of the expression profile toward mature cardiomyocytes [2]. It is evident that differentiation involves the activation of specific signaling pathways that regulate gene expression essential for cardiac development, including the upregulation of cardiac-specific transcription factors like GATA4 and NKx2.5 [9,10]. Additionally, the transformation influences cellular morphology, promoting structural changes characteristic of mature cardiomyocytes, such as the formation of sarcomeres and enhanced contractility. Furthermore, this differentiation process is associated with alterations in metabolic activity as cells transition from glycolytic to oxidative phosphorylation metabolism, crucial for the energy demands of functional heart tissue. Therefore, it is important to note that the NGS data presented in our study are specific only to the undifferentiated state of H9c2 cells. It is now crucial to analyze the mRNA expression profile of H9c2 cells in different cellular states. This analysis has the potential to identify novel pathways that drive the differentiation process.

The absence of *Oct4* expression in H9c2 cells under the chosen culture conditions suggests that they have likely passed the pluripotent stage and are now committed to a differentiated state, which aligns with their origin as embryonic cardiac myoblasts. This is in contrast to induced pluripotent stem cells (iPSCs), where *Oct4* can be reintroduced to convert differentiated cells back to a pluripotent state [64,65,66].

Moreover, analysis of representative cardiac-specific proteins showed that H9c2 cells exhibited a strong expression of Actn1, while Actn2 was only expressed at a low level at the mRNA level and virtually absent at the protein level. Similarly, we found no expression of troponin I (*Tnni3*) at the mRNA and protein levels under the chosen culture conditions. Interestingly, we observed mRNA expression of troponin T (*Tnnt2*) but failed to detect TNNT2 protein expression in H9c2 cells. Conducting a differential expression analysis comparing H9c2 cells to other cardiac or non-cardiac cell lines, or studying gene expression changes throughout differentiation processes, could provide valuable insights into the regulatory pathways controlling cardiomyocyte differentiation and maturation. These comparative studies may help pinpoint key genes and networks involved in cardiac development and diseases. Additionally, future research on DNA methylation patterns could offer further understanding of the epigenetic control of gene expression in H9c2 cells and how they differ from primary cardiac cells. Furthermore, further in-depth studies investigating the expression changes in H9c2 cells in response to various stimuli, such as pro-inflammatory cytokines (IL-1β, IL-6, TNF-α) and endotoxins like lipopolysaccharide (LPS), would be an intriguing avenue for future research. These studies could offer valuable insights into the molecular mechanisms of cardiac inflammation and stress responses, thereby enhancing the utility of H9c2 cells as a model for cardiac disease and therapeutic interventions.

In sum, while H9c2 cells are widely utilized across numerous studies within cardiovascular research, with over 7000 publications citing their use, e.g., [67,68,69], there remains a need for ongoing validation and characterization efforts among different laboratories to ensure consistency in results across various experimental settings. Therefore, we encourage researchers utilizing this cell line to consider both its strengths and limitations carefully while exploring innovative approaches to enhance its applicability within cardiovascular medicine. Future investigations focusing on gene editing techniques or co-culture systems with primary cardiomyocytes could further elucidate mechanisms underlying cardiac function and disease processes using H9c2-derived models.

## 5. Conclusions

In conclusion, this study provides a thorough genetic characterization of the H9c2 cell line, which serves as an important model for cardiac myoblast research. By establishing a comprehensive short tandem repeat (STR) profile and karyotype, we have confirmed the identity and stability of H9c2 cells, addressing concerns regarding genetic drift that may affect experimental reproducibility. The use of NGS has elucidated a diverse gene expression profile that supports the cardiac lineage of H9c2 cells, revealing key markers associated with cardiomyocyte functionality and development. H9c2 cells exhibit significant morphological and biochemical characteristics typical of cardiac myoblasts, including the expression of essential proteins involved in muscle contraction and cytoskeletal organization. While acknowledging their limitations as an in vitro model compared to primary cardiomyocytes, our work highlights the relevance of H9c2 cells in cardiovascular research. The established genetic resources from this study will not only facilitate future investigations into cardiac function but also support the ongoing efforts to develop therapeutic strategies for heart-related conditions and to use H9c2 cells in research aiming to understand the complexities of cardiac biology and pathology.

## Figures and Tables

**Figure 1 cells-14-00502-f001:**
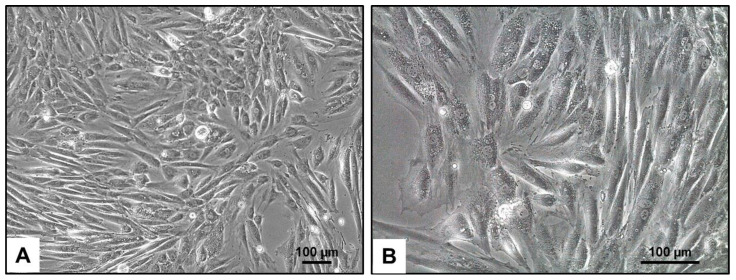
Light microscopic appearance of H9c2 cells. (**A**,**B**) In culture, H9c2 cells typically have a fibroblast-like morphology, with elongated and spindle-shaped cells that can form a confluent monolayer. They also show a high degree of adherence to the culture substrate, allowing them to grow in a dense configuration. The magnifications are as follows: 100× (**A**) and 200× (**B**). The scale bars represent 100 µm.

**Figure 2 cells-14-00502-f002:**
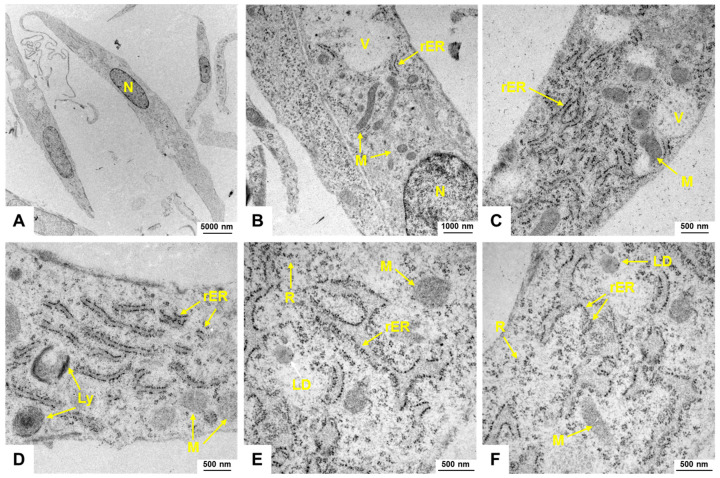
Electron microscopic appearance of H9c2 cells. (**A**–**F**) H9c2 cells display an elongated spindle-shaped morphology with a centrally located nucleus (N). Mitochondria (M) are visible as elongated or spherical organelles with double membranes and cristae. The cytoplasm contains lipid droplets (LD), lysosomes (Ly), ribosomes (R), and vesicles (V). The endoplasmic reticulum (rER) of H9c2 cells forms a network of membranous tubules and flattened sacs, with rough endoplasmic reticulum identifiable by ribosomes on its surface. The images were captured at (**A**) 2156×, (**B**) 10,000×, (**C**,**D**) 21,560×, and (**E**,**F**) 27,800×, respectively.

**Figure 3 cells-14-00502-f003:**
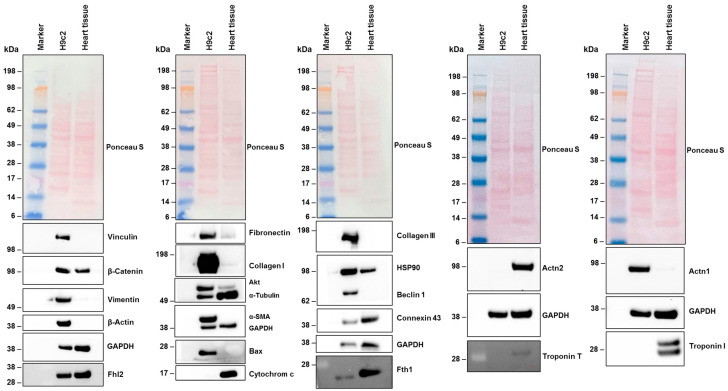
Protein expression in H9c2 cells. Cell protein extracts were prepared from H9c2 cells and rat heart tissue. The proteins (50 µg protein per lane) were then analyzed by Western blot to determine the expression of specific proteins. Ponceau S staining and probing with a glyceraldehyde 3-phosphate dehydrogenase (GAPDH)-specific antibody were used as controls to ensure equal protein loading. Size markers are indicated on the left margin of each Western blot.

**Figure 4 cells-14-00502-f004:**
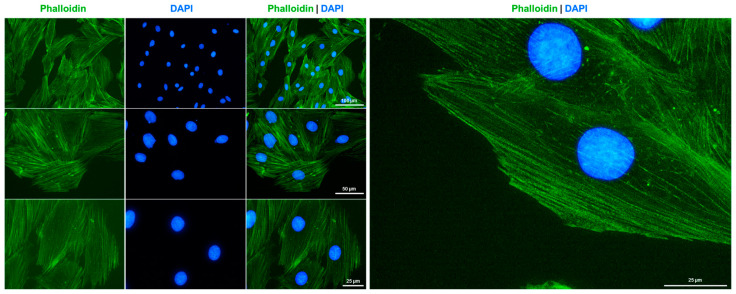
F-actin cytoskeleton staining in H9c2 cells. The cytoskeleton of cultured H9c2 cells was labeled with Alexa Fluor 488^TM^-labeled Phalloidin conjugate (green) and nuclei were counterstained with DAPI (blue). Images were captured using a Nikon Eclipse E80i fluorescence microscope at 200×, 400×, or 600× magnification. Scale bars at the different magnifications are indicated in the images.

**Figure 5 cells-14-00502-f005:**
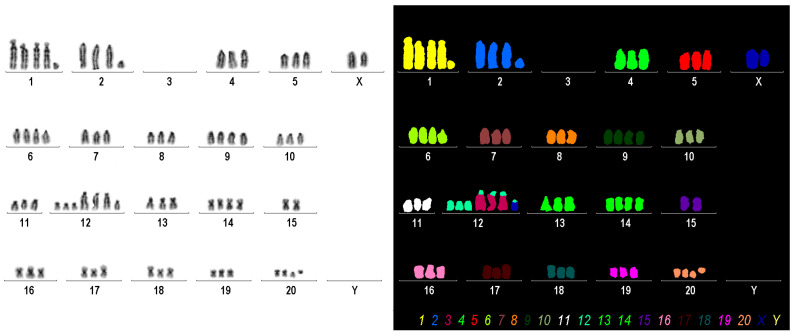
Karyogram analysis and mFISH results of the H9c2 cell line. The left panel displays a representative image of inverted DAPI-banding. The right panel shows a representative mFISH result of the same metaphase, generated using the commercially available 22xRat probe. In this analysis, interchromosomal rearrangements in H9c2 chromosomes are visible as color changes within single chromosomes. The color code for each chromosome is provided at the bottom of this panel. Evaluation of this analysis, according to the International System for Human Cytogenomic Nomenclature (ISCN) nomenclature, revealed the following karyotype [52,53]: 74-81<3n>,X,del(X)(q2?q3?),+1,+del(1)(q?12),del(2)(q?13),−3,−3,−3,+6,+9,+der(12)t(3;12)(q11;q1?2)×3,+der(12)(X;12)(q?3;q1?2),+14,del(20)(q10),+del(20)(p10)[cp12].

**Figure 6 cells-14-00502-f006:**
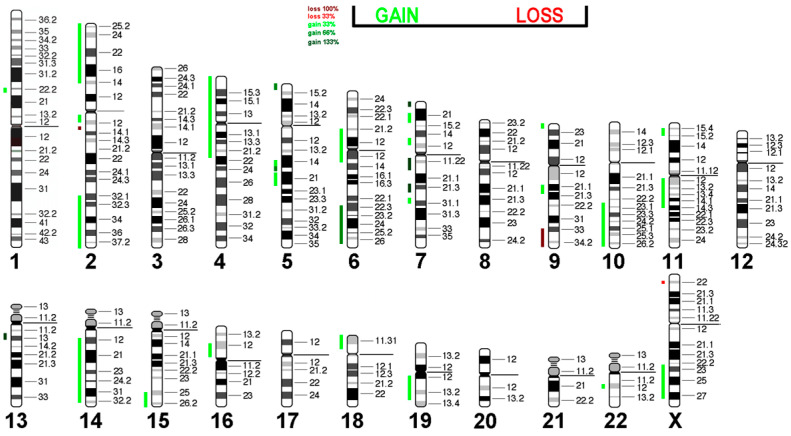
Virtual Comparative Genomic Hybridization results for the H9c2 cell line, translated into the human genome. Copy number alterations are depicted using a color code, with shades of red representing losses and green indicating gains.

**Table 1 cells-14-00502-t001:** Primary and secondary antibodies used for Western blot analysis, listed in alphabetical order ^1^.

Antibody	Cat. no.	Company	Dilution	Clonality
α-Actinin 1	6487	Cell Signaling Technology, Leiden, The Netherlands	1:1000	r mAb
α-Actinin 2	A7811	Sigma-Aldrich, Merck, Taufkirchen, Germany	1:10,000	m mAb
α-SMA	CBL171-I	Sigma-Aldrich, Merck, Taufkirchen, Germany	1:1000	m mAb
α-Tubulin (B-7)	sc-5286	Santa Cruz, Santa Cruz, CA, USA	1:1000	m mAb
β-Actin	A5441	Sigma-Aldrich, Merck, Taufkirchen, Germany	1:10,000	m mAb
β-Catenin (E-5)	sc-7963	Santa Cruz, Santa Cruz, CA, USA	1:1000	m mAb
Acsl1	4047S	Cell Signaling Technology, Leiden, The Netherlands	1:1000	r pAb
Akt (pan) (11E7)	4685	Cell Signaling Technology, Leiden, The Netherlands	1:1000	r mAb
Bax	2772S	Cell Signaling Technology, Leiden, The Netherlands	1:1000	r pAb
Beclin-1 (D40C5)	3495	Cell Signaling Technology, Leiden, The Netherlands	1:1000	r mAb
Collagen I	14695-1-AP	Proteintech, Chromo Tek GmbH, Planegg-Martinsried, Germany	1:1000	r pAb
Collagen III	22734-1-AP	Proteintech, Chromo Tek GmbH, Planegg-Martinsried, Germany	1:1000	r pAb
Connexin 43 (C-20)	sc-6560-R	Santa Cruz, Santa Cruz, CA, USA	1:1000	r pAb
Cytochrome c (D18C7)	11940	Cell Signaling Technology, Leiden, The Netherlands	1:1000	r mAb
Ferritin heavy chain (B-12)	sc-376594	Santa Cruz, Santa Cruz, CA, USA	1:500	m mAb
Fhl2	AF4758	R&D Systems, Bio-Techne, Abingdon, UK	1:500	g pAb
Fibronectin	AB1954	Sigma-Aldrich, Merck, Taufkirchen, Germany	1:3000	r pAb
GAPDH (6C5)	sc-32233	Santa Cruz, Santa Cruz, CA, USA	1:1000	m mAb
HSP90 (C45G5)	4877	Cell Signaling Technology, Leiden, The Netherlands	1:1000	r mAb
Troponin I (cardiac)	ab209809	Abcam, Cambridge, UK	1:1000	r mAb
Troponin T (cardiac)	5593	Cell Signaling Technology, Leiden, The Netherlands	1:1000	r pAb
Vimentin	ab92547	Abcam, Cambridge, UK	1:3000	r mAb
Vinculin	66305-1-Ig	Proteintech, Chromo Tek GmbH, Planegg-Martinsried, Germany	1:5000	m mAb
Goat anti-rabbit IgG (H+L), HRP	31460	Invitrogen, Thermo Fisher Scientific, Schwerte, Germany	1:5000	g
Goat anti-mouse IgG (H+L), HRP	31430	Invitrogen, Thermo Fisher Scientific, Schwerte, Germany	1:5000	r
Mouse anti-goat IgG (H+L), HRP	31400	Invitrogen, Thermo Fisher Scientific, Schwerte, Germany	1:5000	m

^1^ Abbreviations: g, goat; m, mouse; mAb, monoclonal antibody; pAb, polyclonal antibody; r, rabbit.

**Table 2 cells-14-00502-t002:** Myosin gene expression in H9c2 cells supporting their cardiac origin.

Transcript Id ^1^	Gene Id	Gene	Gene Description	TPM
ENSRNOT00000018097 ENSRNOT00000119402 ENSRNOT00000112334 ENSRNOT00000017838 ENSRNOT00000119402	ENSRNOG00000013262	*Myl1*	myosin light chain 1	5.163359 0.648313 1.223487 1.103736 0.648313
ENSRNOT00000072480 ENSRNOT00000101539	ENSRNOG00000050675	*Myl4*	myosin light chain 4	42.672105 0.14434
ENSRNOT00000089074 ENSRNOT00000099810 ENSRNOT00000082518 ENSRNOT00000094248 ENSRNOT00000107790 ENSRNOT00000085644	ENSRNOG00000054140	*Myl6*	myosin light chain 6	1846.647426 874.627113 25.302537 24.381243 19.012288 11.635463
ENSRNOT00000048453	ENSRNOG00000028837	*LOC120093525*	myosin light chain 6B	43.718117
ENSRNOT00000027445	ENSRNOG00000020246	*Myl9*	myosin light chain 9	1155.386796
ENSRNOT00000023944 ENSRNOT00000112797	ENSRNOG00000017645	*Myl11*	myosin light chain 11	285.690465 0.824397
ENSRNOT00000021048 ENSRNOT00000111518 ENSRNOT00000105340 ENSRNOT00000117541 ENSRNOT00000108694 ENSRNOT00000048125	ENSRNOG00000015278	*Myl12b*	myosin light chain 12B	624.478983 396.879829 166.614506 95.549881 0.898159 0.864107
ENSRNOT00000004236	ENSRNOG00000065740	*Myh2*	myosin heavy chain 2	0.672346
ENSRNOT00000004147 ENSRNOT00000115161	ENSRNOG00000046276	*Myh3*	myosin heavy chain 3	103.894988 0.387361
ENSRNOT00000004295 ENSRNOT00000082871 ENSRNOT00000045718	ENSRNOG00000049695	*Myh4*	myosin heavy chain 4	33.85847 0.142894 0.086715
ENSRNOT00000115198	ENSRNOG00000025757	*Myh6*	myosin heavy chain 6	0.050162
ENSRNOT00000024186	ENSRNOG00000016983	*Myh7*	myosin heavy chain 7	0.048688
ENSRNOT00000025859	ENSRNOG00000018997	*Myh7b*	myosin heavy chain 7B	0.608836
ENSRNOT00000105953	ENSRNOG00000068010	*Myh8*	myosin heavy chain 8	0.080643
ENSRNOT00000037611 ENSRNOT00000007398 ENSRNOT00000116925 ENSRNOT00000119854	ENSRNOG00000049236	*Myh9*	myosin heavy chain 9	859.802126 397.860041 10.572224 0.960109
ENSRNOT00000105761 ENSRNOT00000113616 ENSRNOT00000065895	ENSRNOG00000002886	*Myh10*	myosin heavy chain 10	83.625491 31.943467 19.761232
ENSRNOT00000084608 ENSRNOT00000112644	ENSRNOG00000057880	*Myh11*	myosin heavy chain 11	4.104926 1.264626
ENSRNOT00000118051	ENSRNOG00000067378	*Myh13*	myosin heavy chain 13	0.010706
ENSRNOT00000091760	ENSRNOG00000020014	*Myh14*	myosin heavy chain 14	0.04009
ENSRNOT00000090307	ENSRNOG00000061038	*Myh15*	myosin heavy chain 15	1.364068
ENSRNOT00000106031	ENSRNOG00000004177	*Myo1a*	myosin IA	0.158265
ENSRNOT00000068433 ENSRNOT00000108379 ENSRNOT00000118845	ENSRNOG00000048152	*Myo1b*	myosin Ib	1.012463 0.817926 0.124786
ENSRNOT00000117022 ENSRNOT00000036666 ENSRNOT00000108756 ENSRNOT00000101210	ENSRNOG00000004072	*Myo1c*	myosin 1C	256.059864 57.901436 15.070173 5.120976
ENSRNOT00000004609 ENSRNOT00000108255	ENSRNOG00000003276	*Myo1d*	myosin ID	92.963065 0.537064
ENSRNOT00000104863	ENSRNOG00000061928	*Myo1e*	myosin IE	56.688608
ENSRNOT00000011513 ENSRNOT00000110236	ENSRNOG00000008409	*Myo1f*	myosin IF	1.427103 0.040403
ENSRNOT00000119331	ENSRNOG00000059140	*Myo1g*	myosin IG	0.101252
ENSRNOT00000078807	ENSRNOG00000047191	*Myo1h*	myosin IH	0.037109
ENSRNOT00000109142 ENSRNOT00000082288 ENSRNOT00000102636 ENSRNOT00000091789	ENSRNOG00000058866	*Myo5a*	myosin VA	25.016567 9.28073 8.944404 0.299799
ENSRNOT00000094389 ENSRNOT00000019512	ENSRNOG00000014104	*Myo5b*	myosin Vb	5.440433 1.151928
ENSRNOT00000120263 ENSRNOT00000108142 ENSRNOT00000112254	ENSRNOG00000011852	*Myo6*	myosin VI	25.109277 1.384369 0.20236
ENSRNOT00000019053 ENSRNOT00000103282	ENSRNOG00000013641	*Myo7a*	myosin VIIA	26.382815 11.975166
ENSRNOT00000046864	ENSRNOG00000015035	*Myo7b*	myosin VIIb	0.009858
ENSRNOT00000104304 ENSRNOT00000103923 ENSRNOT00000015963 ENSRNOT00000118561	ENSRNOG00000011619	*Myo9a*	myosin IXA	6.65948 4.352163 3.013147 0.953695
ENSRNOT00000045099 ENSRNOT00000083651 ENSRNOT00000081321	ENSRNOG00000016256	*Myo9b*	myosin IXb	16.306978 15.395789 1.48221
ENSRNOT00000065897 ENSRNOT00000102421	ENSRNOG00000010161	*Myo10*	myosin X	61.874959 1.946731
ENSRNOT00000079133	ENSRNOG00000059219	*Myo15a*	myosin XVA	0.017051
ENSRNOT00000035001	ENSRNOG00000042445	*Myo15b*	myosin XVB	0.023935
ENSRNOT00000088919 ENSRNOT00000109794 ENSRNOT00000102240 ENSRNOT00000110549 ENSRNOT00000100376	ENSRNOG00000033101	*Myo18a*	myosin XVIIIa	17.10711 14.92682 13.895175 0.924955 0.563736
ENSRNOT00000098133	ENSRNOG00000048430	*Myo18b*	myosin XVIIIb	0.048917
ENSRNOT00000003886 ENSRNOT00000119574	ENSRNOG00000002852	*Myo19*	myosin XIX	8.602543 1.680903
ENSRNOT00000050443 ENSRNOT00000110793 ENSRNOT00000041328 ENSRNOT00000114924	ENSRNOG00000018630	*LOC108351137*	glyceraldehyde-3-phosphate dehydrogenase	2955.510839 902.439425 147.63458 41.959217

^1^ For comparison of transcript levels of the listed genes, the expression of glyceraldehyde-3-phosphate dehydrogenase (*LOC108351137*) is shown. This gene is known to have consistent expression in the human heart, regardless of the presence of heart failure, and regardless of the specific part of the heart [28]. The complete mRNA expression profile of H9c2 cells observed by NGS can be found in Appendix A. TPM, Transcripts Per Million.

**Table 3 cells-14-00502-t003:** Selected gene expression in H9c2 cells supporting their cardiac origin.

Transcript Id ^1^	Gene Id	Gene	Gene Description	TPM
ENSRNOT00000091560 ENSRNOT00000079824 ENSRNOT00000088795 ENSRNOT00000112260	ENSRNOG00000056756	*Actn1*	actinin, alpha 1	360.869674 121.742056 95.274999 1.456574
ENSRNOT00000101663 ENSRNOT00000101075	ENSRNOG00000017833	*Actn2*	actinin, alpha 2	0.988042 0.866778
ENSRNOT00000011773 ENSRNOT00000116592	ENSRNOG00000008536	*Actc1*	actin, alpha, cardiac muscle 1	29.680643 0.241854
ENSRNOT00000089988	ENSRNOG00000001821	*Adipoq*	adiponectin, C1Q and collagen domain containing	0.088185
ENSRNOT00000005551 ENSRNOT00000119052 ENSRNOT00000102094	ENSRNOG00000004143	*Adipor1*	adiponectin receptor 1	219.61476 0.5584 0.233142
ENSRNOT00000010556	ENSRNOG00000007990	*Adipor2*	adiponectin receptor 2	91.021116
ENSRNOT00000104562 ENSRNOT00000002738	ENSRNOG00000001989	*Alcam*	activated leukocyte cell adhesion molecule	34.73353 10.607857
ENSRNOT00000108356 ENSRNOT00000025258 ENSRNOT00000120144 ENSRNOT00000097402 ENSRNOT00000117221 ENSRNOT00000108831	ENSRNOG00000018598	*Ankrd1*	ankyrin repeat domain 1	1099.57145 696.594054 536.462968 127.375767 125.63322 0.202132
ENSRNOT00000024347 ENSRNOT00000001738	ENSRNOG00000001285	*Atp2a2*	ATPase sarcoplasmic/endoplasmic reticulum Ca^2+^ transporting 2	163.17185 128.264822
ENSRNOT00000019554 ENSRNOT00000111490 ENSRNOT00000109811 ENSRNOT00000108710 ENSRNOT00000113833 ENSRNOT00000116052	ENSRNOG00000014453	*Anxa5*	annexin A5	326.883008 326.883008 7.961338 1.032791 0.881536 0.076392
ENSRNOT00000106165 ENSRNOT00000111975 ENSRNOT00000096029 ENSRNOT00000014464 ENSRNOT00000119845	ENSRNOG00000010668	*Anxa6*	annexin A6	292.28195 104.941243 68.292397 2.736565 1.174227
ENSRNOT00000073636 ENSRNOT00000080190	ENSRNOG00000047466	*Bdnf*	brain-derived neurotrophic factor	22.063067 2.770632
ENSRNOT00000083268 ENSRNOT00000012957	ENSRNOG00000009694	*Bmp4*	bone morphogenetic protein 4	41.067185 28.953671
ENSRNOT00000080271	ENSRNOG00000057713	*Cav2*	caveolin 2	38.392081
ENSRNOT00000007601	ENSRNOG00000005798	*Cav3*	caveolin 3	35.815516
ENSRNOT00000115561	ENSRNOG00000015602	*Cdh2*	cadherin 2	0.034055
ENSRNOT00000019652	ENSRNOG00000014254	*Cpt1a*	carnitine palmitoyltransferase 1A	21.367628
ENSRNOT00000080598 ENSRNOT00000067011 ENSRNOT00000097951	ENSRNOG00000003772	*Csrp2*	cysteine and glycine-rich protein 2	46.077184 15.957459 5.717219
ENSRNOT00000101345 ENSRNOT00000079085	ENSRNOG00000054172	*Ctnnb1*	catenin beta 1	148.736322 133.524514
ENSRNOT00000026860	ENSRNOG00000019810	*Des*	desmin	4.863506
ENSRNOT00000081061 ENSRNOT00000034372 ENSRNOT00000094721 ENSRNOT00000091467 ENSRNOT00000109716	ENSRNOG00000046366	*Dmd*	dystrophin	5.83409 4.877158 4.242989 3.755658 2.561635
ENSRNOT00000005612 ENSRNOT00000118138 ENSRNOT00000096738	ENSRNOG00000004078	*Eno3*	enolase 3	33.28236 24.245848 0.551336
ENSRNOT00000017325	ENSRNOG00000012879	*Fabp3* *(H-Fabp)*	fatty acid binding protein 3	13.928079
ENSRNOT00000023388	ENSRNOG00000017392	*Fgf2*	fibroblast growth factor 2	1.080419
ENSRNOT00000114387 ENSRNOT00000101412 ENSRNOT00000023014	ENSRNOG00000016866	*Fhl2*	four and a half LIM domains 2	77.911289 59.661397 45.106886
ENSRNOT00000014320	ENSRNOG00000010708	*Gata4*	GATA binding protein 4	4.261875
ENSRNOT00000081399	ENSRNOG00000023433	*Gata6*	GATA binding protein 6	11.70926
ENSRNOT00000001054 ENSRNOT00000100494	ENSRNOG00000000805	*Gja1*	gap junction protein, alpha 1	99.918567 17.905309
ENSRNOT00000079552	ENSRNOG00000060448	*Hand2* *(dHand)*	heart and neural crest derivatives expressed 2	70.427748
ENSRNOT00000118307 ENSRNOT00000085680 ENSRNOT00000005995	ENSRNOG00000004517	*Igf1*	insulin-like growth factor 1	22.187026 1.599373 1.449521
ENSRNOT00000020885 ENSRNOT00000117668	ENSRNOG00000015068	*Il11ra*	interleukin 11 receptor subunit alpha 1	28.250421 0.130904
ENSRNOT00000019844	ENSRNOG00000014426	*Lox*	lysyl oxidase	279.086949
ENSRNOT00000076230 ENSRNOT00000076992 ENSRNOT00000076481 ENSRNOT00000075931 ENSRNOT00000076136	ENSRNOG00000033134	*Mef2c*	myocyte enhancer factor 2C	21.461792 10.307472 2.878599 2.158897 0.445899
ENSRNOT00000055680	ENSRNOG00000046424	*Mfn2*	mitofusin 2	65.615575
ENSRNOT00000051121	ENSRNOG00000008658	*Mitf*	Melanocyte inducing transcription factor	3.362071
ENSRNOT00000042686	ENSRNOG00000031093	Mov10l1	Mov10 like RISC complex RNA helicase 1	0.030013
ENSRNOT00000028155	ENSRNOG00000020747	*Nkx2-5*	NK2 homeobox 5	0.319419
ENSRNOT00000026212 ENSRNOT00000104296	ENSRNOG00000019322	*Notch1*	notch receptor 1	8.838129 0.095737
ENSRNOT00000090547 ENSRNOT00000102686	ENSRNOG00000054212	*Pde1a*	phosphodiesterase 1A	1.752889 0.660082
ENSRNOT00000111781 ENSRNOT00000101684 ENSRNOT00000113369 ENSRNOT00000066384 ENSRNOT00000110594 ENSRNOT00000112056	ENSRNOG00000042536	*Pde4d*	phosphodiesterase 4D	9.706262 7.996515 2.912791 1.736483 0.588956 0.028455
ENSRNOT00000073486	ENSRNOG00000046057	*Pnmt*	phenylethanolamine-N-methyltransferase	0.161158
ENSRNOT00000028636	ENSRNOG00000021090	*Pygm*	glycogen phosphorylase, muscle associated	20.312461
ENSRNOT00000008319 ENSRNOT00000075962 ENSRNOT00000067543 ENSRNOT00000091249	ENSRNOG00000005906	*Rgd1565355 (CD36)*	similar to fatty acid translocase/CD36	0.680231 0.346115 0.162213 0.109615
ENSRNOT00000111439	ENSRNOG00000017060	*Ryr2*	ryanodine receptor 2	0.00405
ENSRNOT00000084391	ENSRNOG00000000565	*Sgpl1*	sphingosine-1-phosphate lyase 1	122.518127
ENSRNOT00000025606 ENSRNOT00000094646	ENSRNOG00000018943	*Tnnc1*	troponin C1, slow skeletal and cardiac type	140.282686 69.212851
ENSRNOT00000110513	ENSRNOG00000018250	*Tnni3*	troponin I3, cardiac type	0.528838
ENSRNOT00000034957 ENSRNOT00000058843	ENSRNOG00000028041	*Tnnt1*	troponin T1, slow skeletal type	86.297291 4.163693
ENSRNOT00000084986 ENSRNOT00000108522 ENSRNOT00000047682 ENSRNOT00000050284	ENSRNOG00000033734	*Tnnt2*	troponin T2, cardiac type	385.200438 161.390324 130.206578 6.04521
ENSRNOT00000057641 ENSRNOT00000024575 ENSRNOT00000048044 ENSRNOT00000090288 ENSRNOT00000085894 ENSRNOT00000024617 ENSRNOT00000099012 ENSRNOT00000040808 ENSRNOT00000112475 ENSRNOT00000024493	ENSRNOG00000018184	*Tpm1*	tropomyosin 1	609.160335 394.939461 348.84129 309.385315 250.542706 70.487194 62.960197 23.306842 9.964889 9.432292
ENSRNOT00000026493	ENSRNOG00000019486	*Trpv1*	transient receptor potential cation channel, subfamily V, member 1	0.210286
ENSRNOT00000108121 ENSRNOT00000101577 ENSRNOT00000107188 ENSRNOT00000114553	ENSRNOG00000069271	*Ttn*	titin	2.315625 1.775349 0.597153 0.459324
ENSRNOT00000027487 ENSRNOT00000076187	ENSRNOG00000020276	*Tnnt2*	troponin I2, fast skeletal type	10.186619 0.503748
ENSRNOT00000014127 ENSRNOT00000079275	ENSRNOG00000010390	*Hmbs*	hydroxymethylbilane synthase	27.1177 0.615926
ENSRNOT00000096774 ENSRNOT00000035628	ENSRNOG00000008195	*Ywhaz*	tyrosine 3-monooxygenase/tryptophan 5-monooxygenase activation protein, zeta	529.169941 197.545795

^1^ For comparison of transcript levels of the listed genes, the expression of hydroxymethylbilane synthase (*Hbms*) and tyrosine 3-monooxygenase/tryptophan 5-monooxygenase activation protein, zeta (*Ywhaz*) are depicted. These genes are frequently used as housekeeping genes in gene expression studies in human cardiomyoctes [28]. The complete mRNA expression profile of H9c2 cells observed by NGS can be found in Appendix A. TPM, Transcripts Per Million.

**Table 4 cells-14-00502-t004:** Losses and gains of chromosomal regions in H9c2 cells based on mFISH and inverted DAPI-banding analyses ^1^.

Rat RGSC 5.0/rn5		Human GRCh37/hg19
Gain		
1pter->1q?12 (+2) => 1pter->1q12	1-75515806	6q22.31q27 5pterp15.31 5q15q15
1q12->1qter (+1)	75515807-290094216	10q23.2q26.3 11q12.1q14.3 15q25.1qter 11p15.4p15.2 19q12q13.33 9q21.11q21.31 9pterp24.1 16p12.3p11.2
2pter->q?13 => (+1) 2pter->2q13	1-40882813	5q14.1q15
6pter->6qter (+1)	1-156897508	14q12q32.33 2pterp16.3 7p21.2p15.3 7q22.3q31.1
9pter->9qter (+1)	1-121549591	2q32.1qter 6p21.2q13 2q11.2q12.2 18p11.32p11.22 5q21.1q22.1
+12pter->q1?2 (+4) =12pter->q12	1-31247709	13q12.13q13.2 7pterp22.1 7q11.21q11.22 7q21.2q22.1 7q11.23q11.23
14pter->14qter (+1)	1-115151701	4pter->q22.1 2p16.2p14 7p13p12.1 22q12.1q12.3 1p22.2p22.1
Xq3?->Xqter (+1) => Xq34->Xqter	111295344-154597545	Xq22.3q37.3
Loss		
3pter->3q11 (−3)	1-22900371	9q33.2q34.3 2q13q13
Xq2?->Xq3? (−1) => Xq22->Xq34	57322686-111295344	Xp22.11q22.3

^1^ For each region, the cytogenetic span and approximate molecular span in the rat genome are provided, along with the approximate cytogenetic span when projected onto the human genome. In cases where breakpoints were not exactly determined, they were transformed into the most likely regions of copy number alterations (CNAs) as highlighted by a => number of gains or losses indicated in curly brackets. The question mark (?) in karyotype analysis indicates uncertainty in identifying a chromosome or chromosome structure. In this case, the breakpoint region cannot be definitively defined.

**Table 5 cells-14-00502-t005:** STR-based DNA profiling of H9c2 cells using 31 species-specific STR markers.

			Allele Sizes (bp) in				
SN	Marker Name ^1^	Location on Chromosome	H9c2	CFSC-2G	HSC-T6	PAV-1	Rat-1
1	73	1	211	194, 203	194	194	211, 213
2	8	2	195	236	234	234, 238	232, 236
3	2	2	127	126	127	128	129
4	4	3	250, 252	268, 270	238	236, 238	250, 252
5	3	3	160	160, 182	160, 162	162	178, 182
6	26	4	148	150	166	154	162
7	19	4	174	180	175	179	176, 178
8	81	5	127	130, 134	130, 132	130	128
9	34	6	193	184, 189	188	182, 187	184, 189
10	30	7	192, 194	188, 192	186, 192	192	186
11	24	8	258, 260	260	249, 253	254, 259	247, 249
12	59	9	143, 145	145	143, 146, 180	145, 148	176, 178
13	62	9	154	166	177	166	154
14	1	10	104	105	96	96, 105	96
15	55	10	205, 207	210, 214	210, 218	210, 218	203, 205
16	36	11	263	222	234	222	228
17	67	11	161	154, 156	165	165	165, 167
18	13	12	121	121	121, 135	121	121
19	35	13	203	197	197, 203	203	203
20	42	13	149	125	127	144, 156	154, 156
21	70	14	175	158, 175	175, 179	158, 175	158
22	61	15	128	128	128	128	110
23	79	15	172	172, 180	172	172	172
24	90	16	172, 174	159, 161	174	175	159, 161
25	69	16	143	138	139	136, 139	148
26	78	17	151, 153	136, 151	147, 151	147, 149	136, 140
27	15	18	232	232	232	232	238
28	16	18	243	251, 260	247, 251	251	247, 251
29	75	19	140	144	144, 184	144, 184	144
30	96	20	240	210	210, 212	210	208, 210
31	91	20	221	221	205, 211	211, 225	219, 221

^1^ Testing was conducted using the CellCheck^TM^ Rat Panel (IDEXX BioAnalytics, Columbia, MO, USA).

## Data Availability

The original contributions presented in this study are included in the article/Appendix A. Further inquiries can be directed to the corresponding authors.

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
