# Peer review of "Genetic and Molecular Characterization of H9c2 Rat Myoblast Cell Line"

_cells, 2025, doi:10.3390/cells14070502_

Round 1
Reviewer 1 Report
Comments and Suggestions for Authors
This study is well-designed and thoroughly executed. Liehr et al. characterized the widely used embryonic rat cell line H9c2 through comprehensive genetic and molecular analyses. They performed karyotyping and short tandem repeat (STR) analysis to confirm the cell line’s identity and stability, utilizing techniques such as STR profiling and mFISH. Additionally, they conducted virtual Comparative Genomic Hybridization (vCGH) to assess genomic alterations, including gains, losses, and structural rearrangements. To further investigate molecular characteristics, they employed Next-Generation Sequencing (NGS) to profile the mRNA expression of H9c2 cells. Finally, they validated their RNA sequencing findings by examining protein abundance and cytoskeletal organization using Western blotting (WB), ICC, and electron microscopy.
Since different teams use their own cell lines, it is essential to characterize them to ensure the validity of downstream analyses. However, this requirement reduces the novelty and significance of the study, as characterization alone does not offer new functional insights. Nevertheless, the demonstrated genetic instability in H9c2 cells raises important concerns regarding the reliability of experimental conclusions derived from this model. Given the potential variability in downstream analyses, this study serves as a critical reference for researchers utilizing H9c2 cells, highlighting the extent of chromosomal instability and urging caution in data interpretation.
Major concerns:
- The title of this study suggests insights into cardiac myoblast function, yet the paper lacks functional analyses such as electromechanical assessments (e.g., mechanical properties or electrophysiological studies). Without these evaluations, there is no direct characterization of the functional properties of this cell line, particularly in the context of cardiac progenitor features. I recommend adjusting the title to more accurately reflect the study’s scope.
- In the analysis of protein abundance and cytoskeletal organization, the primary focus was on metabolic and non-cardiac protein levels. However, given that one of the primary applications of this cell line is as a cardiac progenitor model, it is essential to confirm and quantify the expression of cardiac-specific proteins. To strengthen the findings, the authors should validate their RNA-seq data by assessing the protein expression of key cardiac markers. I strongly recommend performing Western blot and immunocytochemistry for cardiac-related proteins, such as Tnnt2, Actn2, Myh6, and Myh7, to provide a more comprehensive characterization of the cell line.
Minor concerns:
- Line 24: Please adjust the description of the functional characteristics of H9c2 cells to better reflect the available data.
- Line 29: I recommend adding a sentence about the potential chromosomal drift in this cell line immediately after the statement: “Our findings underpin that H9c2 cells are a reliable model for studying cardiac myoblast biology.” While the authors briefly address this concern at the end of the introduction, incorporating it here would provide a more balanced discussion by acknowledging the genetic instability that could affect experimental reproducibility and data interpretation.
- Figure1, add the A and B in the figure as well.
- Line 227: Please introduce the abbreviation for rough endoplasmic reticulum (RER) in this line and ensure its consistent use in the Figure 2 legend (lines 233 and 234).
- Line 252: It would be beneficial to further elaborate on this paragraph: “Furthermore, the differential expression of specific myosin isoforms may provide insights into the regulatory pathways governing cardiomyocyte differentiation and maturation.” For instance, performing differentially expressed gene (DEG) analysis between this cell line and another relevant cell line, or across different timepoints, could offer a deeper understanding of the regulatory mechanisms involved.
- Line 276: I recommend revising this sentence and reference. Given that one of the primary uses of this cell line is to investigate cardiac differentiation and maturation, it would be more appropriate for the authors to highlight that the absence of Oct4, a marker of pluripotency, indicates the differentiation of these cells and representing a developmental trajectory. In contrast, dedifferentiation occurs when iPSC lines are generated from fibroblasts or other cell types by introducing Oct4, which is opposite to that of the H9c2 cells.
- Figure 3, GAPDH is missing as an internal control in left column.
- Line 303: The authors mention the non-contractile cytoskeletal role in electrical and mechanical coupling. I suggest adding contractile-related protein staining as outlined in major concern 2.
- Line 363: Please add the definition of the question mark (?) in the figure legend.
- Line 392: Do the authors have methylation analysis data? If so, including it could enhance this comparison.
- Data Availability: Highly suggest the authors consider uploading their mRNA sequencing data to an online repository, such as the Genotype-Tissue Expression (GTEx) project, to make the data more accessible to the broader research community.
Author Response
Dear Reviewer 1,
I want to express my gratitude for your thorough and thoughtful review of our paper. Your expertise and constructive feedback were invaluable in enhancing the quality of our work.
The suggestions you provided were not only insightful but also highly applicable, allowing us to refine our arguments and strengthen our overall presentation. We have marked all changes made in the revised version of the paper in red for your convenience.
Thank you once again for your support. We are confident that your contributions have significantly improved our research.
Best regards,
Ralf Weiskirchen

Reviewer 2 Report
Comments and Suggestions for Authors
In the article by Liehr et al. the authors determine the expression of proteins and genes and genomic characteristics of embryonic rat H9c2 cells. These cells have been extensively used as an in vitro model of cardiac myoblasts, however a detailed description of gene expression has not been fully investigated. This detailed documentation of gene expression profile and karyotype will be useful to many researchers that use these cells for their studies.
There are several comments that should be addressed that will improve the information from this study.
- It is not clear as to why the RNA sequencing experiment was aligned to the human genome (GRCh39) as the cells are derived from rat. This was stated in the methods section (lines 150-151).
- The authors highlight the expression of the myosin and other well characterized cardiac genes as 2 tables. The genes are annotated from the Transcript Id, and thus the TPM values are listed for each unique transcript. It is not clear as to how useful this data is and a more detailed bioinformatic study should be conducted. The authors should include Gene Ontology (GO) search results to highlight what Biological processes are occurring in these cells. Also GO Molecular Processes and Cellular Components should be included to show whether these also show the expected cardiac biology.
- Can the tables of transcript TPMs be converted to graphs? It may be meaningful to show the expression of these genes and whether it is comparable to another cell type such as a fibroblast or liver cell?
- The next generation sequencing data should be made available through some database.
Author Response
Dear Reviewer 2,
I want to express my gratitude for your thorough and thoughtful review of our paper. Your expertise and constructive feedback were invaluable in enhancing the quality of our work.
The suggestions you provided were not only insightful but also highly applicable, allowing us to refine our arguments and strengthen our overall presentation. We have marked all changes made in the revised version of the paper in red for your convenience.
Thank you once again for your support. We are confident that your contributions have significantly improved our research.
Best regards,
Ralf Weiskirchen

Round 2
Reviewer 1 Report
Comments and Suggestions for Authors
The authors have satisfactorily addressed my concerns.
Reviewer 2 Report
Comments and Suggestions for Authors
The authors have responded to my comments and offered valid reasons for not directly including the suggestions offered.
I agree that their rebuttal do offer valid points. But I think this study could be compared to another cell line that has transcriptomics performed. It would be preferable to compare these to the rat1 fibroblast line, which they have already sequenced and published. See Liehr et. al. Cells 2024 (doi: 10.3390/cells14010021)
As it stands this is a useful report on the rat H9c2 karyotype and gene expression profile.